# Anthropomorphic Characterization of Ankle Joint

**DOI:** 10.3390/bioengineering10101212

**Published:** 2023-10-17

**Authors:** Dinesh Gundapaneni, James T. Tsatalis, Richard T. Laughlin, Tarun Goswami

**Affiliations:** 1Department of Biomedical, Industrial and Human Factors Engineering, Wright State University, Dayton, OH 45435, USA; gundapaneni.2@wright.edu; 2Department of Radiology, Miami Valley Hospital, Dayton, OH 45409, USA; jttsatalis@premierhealth.com; 3Department of Orthopedic Surgery, Sports Medicine and Rehabilitation, Miami Valley Hospital, Dayton, OH 45409, USA; laughlrt@ucmail.uc.edu

**Keywords:** ankle, morphology, 3D modeling, replacement, regression

## Abstract

Even though total ankle replacement has emerged as an alternative treatment to arthrodesis, the long-term clinical results are unsatisfactory. Proper design of the ankle device is required to achieve successful arthroplasty results. Therefore, a quantitative knowledge of the ankle joint is necessary. In this pilot study, imaging data of 22 subjects (with both females and males and across three age groups) was used to measure the morphological parameters of the ankle joint. A total of 40 measurements were collected by creating sections in the sagittal and coronal planes for the tibia and talus. Statistical analyses were performed to compare genders, age groups, and image acquisition techniques used to generate 3D models. About 13 measurements derived for parameters (TiAL, SRTi, TaAL, SRTa, TiW, TaW, and TTL) that are very critical for the implant design showed significant differences (*p*-value < 0.05) between males and females. Young adults showed a significant difference (*p*-value < 0.05) compared to adults for 15 measurements related to critical tibial and talus parameters (TiAL, TiW, TML, TaAL, SRTa, TaW, and TTL), but no significant differences were observed between young adults and older adults, and between adults and older adults for most of the parameters. A positive correlation (r > 0.70) was observed between tibial and talar width values and between the sagittal radius values. When compared with morphological parameters obtained in this study, the sizes of current total ankle replacement devices can only fit a very limited group of people in this study. This pilot study contributes to the comprehensive understanding of the effects of gender and age group on ankle joint morphology and the relationship between tibial and talus parameters that can be used to plan and design ankle devices.

## 1. Introduction

Over the past 15 years, total ankle replacement (TAR) has emerged as an alternative to ankle arthrodesis [1]. When compared to total hip and knee arthroplasty results, the long-term outcomes of ankle replacement are unsatisfactory [2,3]. The current failure rate of TAR devices is about 10–12% over a period of 5 years. Major complications like infection and component loosening are associated with the failure of these devices [4]. Due to unsatisfactory prosthesis design, the clinical results are disappointing for current-generation devices [5]. Morphology of the bones plays a crucial role in the clinical success of relevant joint arthroplasty [5]. Understanding the ankle joint anatomy and anthropomorphic evaluation is essential to designing a patient-specific implant or deriving the best fit size for a patient [6]. This helps in substantially reducing complications, thereby improving the survival rates of these devices. Therefore, quantitative knowledge of ankle joint morphology is crucial [7].

Passive joint kinematics is a result of the complex interaction between the articulation surfaces and ligament constraints [8]. During the stance phase of the gait cycle, most loading on the joint occurs across the articular surfaces, and the stabilization due to ligaments is minimal [9]. It is very important to study the trapezium shape of the talocrural joint since the articulating surface of the joint contributes 70% to anteroposterior stability, 50% to version stability, and 30% to rotational stability [10,11,12,13]. To design a prosthesis, thorough knowledge of joint mobility and stability is required in addition to the geometry of the joint [14]. To perform measurements over a large population, the methods adopted should be consistent and accurate, and the data collected should be reliable [5]. Errors in the estimation could affect pre-surgical decision making, which involves appropriate size selection of the implant [15]. The radius of a component being smaller than normal could result in a slackening of ligaments, whereas a larger component leads to motion constraint [7]. It is crucial to use an appropriately sized component to eliminate the risk of edge loading, and for a better long-term fixation, a shape match between the bony surface (after osteotomy) and the implant surface is necessary [7,16].

Several studies have concluded that the left and right talus bones of intact human ankle joints exhibit bilateral symmetry [17]. This allows the surgeons to use the contralateral unaffected side as a reference template to treat a fracture or degenerative side of the joint by determining the proper implant size and its position [17,18]. In certain cases, the application of a reference template to treat the joint can be challenging when both sides are subjected to complex fractures or severe joint disorders [19,20,21,22]. So, it is very important to determine the relationship between the morphological parameters of the ankle joint. The effect of gender on ankle joint morphology is reported by several studies, where the anterior width, posterior width, and trochlea tali length measurements for the males were significantly higher compared to females [23]. A study by Hongyu C et al. (2020) reported higher values for tibial parameters (anterior–posterior gap, anterior–posterior inclination angle, and maximal tibial thickness) in females compared to males [24]. Age-related variation in the talar morphology was reported by Nozaki et al. (2020), where, for both the females and males, the anteroposterior length of the trochlea and the talar head surface increased while the length of the talar neck decreased with aging [25]. This shows that ankle joint morphology undergoes age-dependent changes, resulting in variations in its shape, thereby affecting the functionality of the joint.

Studies during the early 2000s measured morphological parameters of the ankle complex using planar radiographs [5,26]. This approach limited their studies to two dimensions, thereby estimating the values that are different from true estimations obtained using 3D data [6]. For instance, by using planar radiographs, the wedge shape of the talar dome, which is wider on the anterior side compared to the posterior, cannot be viewed properly, and planar radiography involves more uncertainties and errors while acquiring the imaging data [6,23], whereas 3D imaging like CT can be easily reformatted and every feature of the bone can be visualized [27]. To develop TAR devices reasonably, at least nine morphological parameters are required, where three parameters can only be obtained using 3D data [5,17,26,28,29]. A study by Rathnayaka et al. (2012) observed an average error of 0.15 mm for CT-based models and an average error of 0.23 mm for MRI models when these models were compared to reference models, which were derived using a mechanical contact 3D scanner [30]. But, no significant difference was observed between the CT and MRI models. Moro-oka et al. (2007) conducted kinematic analysis by using CT and MRI models and reported minimal errors [31]. Therefore, MRI data can be utilized in addition to CT data to measure morphological parameters to replicate joint kinematics accurately.

To maintain consistency, techniques that are widely used in previous studies were adopted. Unlike previous studies (used CT data), this study utilized CT and MRI data to analyze the morphology of the ankle joint by developing 3D models. Even though several studies analyzed 3D morphological parameters in the past, they are limited to very few variables [6,8,23,27]. Therefore, in this pilot study, a comprehensive approach was taken to measure 15 morphological parameters for the tibia and talus (a total of 40 measurements were derived from sagittal and coronal sections) to define the ankle joint morphology. This pilot study analyzed these main morphological parameters of the ankle joint using the radiography data of individuals to provide fundamental information for the development of appropriate ankle devices that can fit across genders and age groups. We hypothesize that (a) a significant difference exists between the genders (males and females), (b) no difference exists between the age groups (young adults, adults, and older adults), (c) no difference exists between the CT and MRI models, and (d) a significant relationship exists between the tibia and talus parameters.

## 2. Materials and Methods

Ankle joint data of 22 patients (8—CT and 14—MRI) taken under passive loading conditions were considered for this pilot study. The study population comprises 12 females with a mean age of 41 ± 19 years (range 19–88 years) and 10 males with a mean age of 47 ± 13 years (range 13–58). All the patient details were anonymized, and only the gender and age of the patients were available. Patients with no deformities, contractures, articular degeneration, or ligament injuries were considered. The study population was divided into three groups to obtain the distribution count equally close between the groups (for better statistical power) rather than with a large difference. The age groups include young adults aged below 30 years (Group 1); adults aged between 30 and 50 years (Group 2); and older adults aged above 50 years (Group 3). Patient demographics and protocols used to acquire the imaging data are provided in Table 1 and Table 2, respectively.

### 2.1. Reference Cardinal System

By using Mimics v.19 (Materialise, Leuven, Belgium), 3D models were developed from imaging data, and these models were exported to 3-Matic v.11 software (Materialise, Leuven, Belgium) to measure the morphological parameters [32]. The morphological parameters measured during this study are provided in Table 3.

To compare with previous studies, techniques that are widely used to measure the morphological parameters were adopted. Initially, a reference cardinal system (consisting of sagittal, transverse, and coronal planes) was defined based on talar anatomical landmarks [6,19].

For the sagittal plane, the coordinate system is translated and rotated so that the datum plane transects in the middle of the talar dome [27]. For the transverse plane, the plane is rotated so that its axis is parallel to the superior talar surface. The coronal plane is perpendicular to the sagittal plane, and it is rotated to transect the talar dome in the middle, as shown in Figure 1 [6,27]. By using extrema analysis in 3-Matic, maximal points were identified on the articulation surface of the talus near the condylar region, as shown in Figure 2. On the medial side of the talus, a datum plane was created parallel to the reference sagittal plane passing through the maximal point. Similarly, a datum plane was created between the lateral side of the trochlea tali and the lateral facet, and the plane was rotated to accommodate the lateral shoulder of the trochlea tali [8]. Later, a mid-sagittal (middle) plane was created by taking the average of existing datum planes (medial and lateral), as shown in Figure 2.

### 2.2. Morphometric Evaluation

By using Boolean operations, three sections of the talus (lateral, middle, and medial) were created based on respective planes to measure the morphological parameters. By using the radius tool in 3-Matic, the sagittal radius of the talus (SRTa) was derived from all three sections (lateral, middle, and medial) by using the 3-point method, as shown in Figure 3.

To obtain trochlea tali arc length (TaAL) in the sagittal plane, the distance between the anterior and posterior points of SRTa was measured, as shown in Figure 3. A talar axis in the coronal plane was derived by connecting the centers of medial and lateral circles (SRTa), as shown in Figure 4. A datum plane was created perpendicular to the transverse plane by using the coronal axis. To create talar dome sections in the coronal plane, additional datum planes were created by rotating the reference plane, as shown in Figure 4. A study by Wiewiorski et al. (2012) used 30 degrees to create sections of the talar dome on the anterior and posterior sides in the coronal plane by using a rotation axis that passes through the center of the mid-sagittal circle [27], whereas Siegler et al. (2014) created five equally spaced sections between the anterior and posterior boundaries of the trochlea tali surface by defining an axis that connects the center of two circles on the medial and lateral side in the coronal plane [8]. In our preliminary analysis, we observed that 30 degrees is not sufficient to accommodate the surface of the talar dome for some large-size models to create sections in the coronal plane. So, this study used a different increment size (a multiple of 7.5 degrees to create sections between the two boundaries of the talar dome in the coronal plane) based on the size of the 3D model.

We created three sections (anterior, central, and posterior) of the talus in the coronal plane. For the talar edge angle (α and β), two lines were used on each side (medial and lateral), one adjusted to the talar dome surface and the other adjusted to the malleolus of the talus, as shown in Figure 4 [33]. To measure the talar edge radius (R_l_ and R_m_), a circle was fitted to the talar edge surface in between the talar edge lines, as shown in Figure 4. To calculate the talus dome ratio (TDR), the distance between the highest points on the medial and lateral sides of the talar edge (b) was measured, and the depth of the talar sulcus (a) was determined by measuring the distance between the line fitted to the talar dome surface to the deepest point of the sulcus as shown in Figure 4 [27]. By merging the sagittal and coronal sections, intersection points were derived. Talar width (TaW) was determined by measuring the distance between the medial and lateral intersection points, as shown in Figure 5. Trochlea tali length (TTL) was obtained by measuring the distance between the anterior and posterior intersection points, and the angle of trochlea tali shape (ATTS) was obtained by measuring the angle between the medial and lateral trochlea tali lengths.

### 2.3. Statistical Analyses

To predict tibia morphological parameters based on obtained values for the talus, we need to establish a significant correlation between them by developing a regression equation. So, the talus cardinal system was used as a reference in this study to measure the tibial morphological parameters (see Appendix A). A total of 40 measurements (including the age of the patient) were considered, with 15 main variables for both the tibia and talus. JMP v.11 (SAS Institute, Cary, NC, USA) was used to conduct statistical analysis. For continuous data, testing for normality is an important step to determine the statistical methods (selection of parametric/nonparametric test) for conducting data analysis based on normality status [34]. The normality test compares the scores in the sample to normally distributed scores that have the same mean and standard deviation [35]. So, all the measured parameters were checked for normality using the Shapiro–Wilk test [15]. A *t*-test is a statistical method used to determine if there is a significant difference between the means of two groups and the relation between them. Student’s *t*-test was used to compare the two gender groups, three age groups, and two image acquisition methods for normally distributed parameters, and the Wilcoxon rank sum test was used for parameters that are non-normal [7]. The paired *t*-test (Matched pairs method) was used to compare the differences between the parameters obtained in different sections for the same variable. To quantify the strength of the relationship between two morphological parameters, a correlation analysis was performed. For normally distributed parameters, Pearson’s correlation coefficient (r) was used, and for non-normal data, Spearman correlation (ρ) was used. The regression analysis (Bivariate) was performed to establish a relationship between the tibia and talus parameters. To deduce how the current TAR devices fit the population in this study, the interquartile range (IQR) is determined by comparing the obtained morphological results with the design parameters of TAR devices by generating the box-and-whisker plot.

## 3. Results

The sample group consisted of 12 females and 10 males with a mean age of 41 ± 19 and 47 ± 13 years, respectively. Due to the wider age range (13–88 years) and limited sample size (*n* = 22), the resultant standard deviation is high. This was also reflected in morphological parameters obtained for the tibia and talus since the size/shape of the bones varies from one person to another. A summary of the obtained results is provided in Table A1 (see Appendix B). The Shapiro–Wilk test results show that out of 40 measured parameters, only 12 were found to be not normally distributed (Age—male; SRTi—medial and lateral; TaAL—middle; TTL—lateral; α—central; β—posterior; R_l_—anterior and posterior; R_m_—anterior and posterior; and TDR—posterior). When compared with females, males showed higher mean values for most of the parameters except the TTL angle, TDR central and posterior, α central and posterior, and β posterior. Only 13 parameters showed a significant difference between males and females based on Student’s *t*-test and Wilcoxon rank sum test results, as shown in Table A1 (see Appendix B). In most cases, the mean values obtained for Group 2 are higher compared to Groups 1 and 3. Similarly, Group 3 has higher mean values compared to Group 1. We observed 15 parameters that showed significant differences between Groups 1 and 2, and only 3 parameters between Groups 1 and 3 (Age, TTL Lateral, and TDR posterior) and between Groups 2 and 3 (Age, TDR Central, and TDR Posterior). Results obtained from Student’s *t*-test and Wilcoxon rank sum test showed no significant difference between the image acquisition methods (CT and MRI) for most parameters, except for the talar edge angles (α—posterior, β—anterior, and posterior) and radius values (R_l_, R_m_—central, and posterior). The tibial sagittal radius (SRTi) averaged 26 ± 9 mm at the medial section, 26 ± 7 mm at the middle section, and 25 ± 5 mm at the lateral section. Similarly, the sagittal radius of the talus (SRTa) averaged 23 ± 6 mm at the medial section, 23 ± 6 mm at the middle section, and 21 ± 4 mm at the lateral section. In both cases, the sagittal radius values of the tibia (SRTi) and talus (SRTa) were decreased linearly from the medial to lateral section. The tibial width (TiW) averaged 27 ± 8 mm at the anterior section, 25 ± 7 mm at the central section, and 24 ± 7 mm at the posterior section. The same trend was observed with talar width (TaW) values averaging 27 ± 6 mm at the anterior section, 24 ± 6 mm at the central section, and 21 ± 6 mm at the posterior section. In both cases, the tibial (TiW) and talar (TaW) width values decreased linearly from the anterior to the posterior section. The length of the tibial mortise (TML) decreased from the medial (25 ± 8 mm) to the lateral section (24 ± 5 mm). Similarly, the length of the trochlea tali (TTL) decreased from the medial (35 ± 8 mm) to the lateral section (32 ± 8 mm). The lateral talar edge radius (R_l_) decreased from the anterior (3 ± 1 mm) to the central (3 ± 2 mm) section, and then it increased to 5 ± 3 mm at the posterior section. Whereas the medial talar edge radius (R_m_) increased from the anterior (4 ± 2 mm) to the central (5 ± 3 mm) section, and then it decreased to 4 ± 2 mm at the posterior section. Both lateral (α) and medial (β) talar edge angles decreased from the anterior (121 ± 19 deg, 122 ± 25 deg) to the central section (101 ± 9 deg, 114 ± 19 deg) and then increased from the central to the posterior (127 ± 20 deg, 116 ± 20 deg) section, respectively. The talar dome ratio (TDR) decreased from the anterior (0.07 ± 0.04) to the posterior section (0.03 ± 0.03).

Based on the paired *t*-test results, it was observed that TiAL middle showed a significant difference with TiAL medial (*p*-value—0.0003) and TiAL lateral (*p*-value—0.01). But, no significant difference was observed between lateral and medial values for TiAL. No significant difference was observed between the SRTi values obtained for different sections. For tibial width (TiW), a significant difference was observed between the values obtained at the anterior, central, and posterior locations (*p*-value—<0.01). No difference was observed between the tibial mortise lengths (TML) obtained at the medial and lateral sections. A significant difference was observed between the lateral and medial values of TaAL (*p*-value of 0.003) and the lateral and middle values (*p*-value of 0.03) for the paired *t*-test. But, no significant difference was observed between the TaAL values obtained at the medial and middle sections. No significant difference between the medial and middle values was observed for SRTa. But, a significant difference between the lateral and medial (*p*-value—0.01) and the lateral and middle was observed (*p*-value—0.002) for SRTa. A significant difference was observed between TaW values obtained at the anterior, central, and posterior sections with a *p*-value < 0.0001, and a significant difference was observed between the trochlear tali lengths (TTL) obtained at the medial and lateral sections (*p*-value of 0.002). A significant difference in talar dome ratios (TDR) was observed between the anterior and central sections (*p*-value of <0.01) and between values obtained at the anterior and posterior sections (*p*-value of <0.01). But, no significant difference was observed between the values obtained at the central and posterior sections for the paired *t*-test.

Radius values obtained for the tibia (SRTi) and talus (SRTa) in different sections (medial, middle, and lateral) were compared, respectively. A significant difference (*p*-value < 0.01) in these values was observed for Bivariate analysis. Comparison between the tibia (TiAL) and talus (TaAL) arc lengths for respective sections showed a significant difference (*p*-value < 0.0001). Only posterior tibial width (TiW) showed a significant difference with posterior talar width (TaW), but no significant difference was observed between the width values obtained in other sections (anterior and central). In the medial section, tibial mortise length (TML) showed a significant difference with the trochlea tali length (TTL) with a *p*-value < 0.0001, but no significant difference was observed between these values in the lateral section. For the Bivariate analysis test results, no significant difference was observed between ATMS and ATTS values. The highest correlations with significance were observed for TaW anterior and TaW posterior (r = 0.88), TaW central and TaW anterior (r = 0.97), TaW posterior and TaW central (r = 0.96), SRTa middle and SRTi middle (r = 0.96), TiW anterior and TiW posterior (r = 0.80), TiW central and TiW anterior (r = 0.95), TiW posterior and TiW central (r = 0.94), and TTL medial and TML medial (r = 0.92). A very low correlation was observed for ATMS and TiAL medial (r = 0.0036). A significant negative correlation was observed for ATTS and TaW posterior (r = −0.76) and β central and ATTS (r = −0.73) (see Appendix A).

Bivariate analysis was used to determine the relationship between the tibia and talus parameters that belong to a similar category (SRTa-SRTi, TaW-TiW, TTL-TML, and ATTS-ATMS). A linear fit was used to generate regression equations (see Appendix A) between these parameters, and corresponding details are provided in Table 4. In all cases, a significant relationship (*p*-value < 0.05) was observed between the tibia and talus parameters. A significant linear relationship was also observed between other parameters of the tibia and talus (SRTa-TaW, TTL, and SRTi-TiW, TML) in all sections, respectively. R-squared values were derived to indicate the goodness of the regression fit, and to further evaluate these values, residual versus fitted values were plotted (see Appendix A). In all cases, the residuals showed random patterns, indicating a good linear fit for the model.

Size comparisons were made by generating the box-and-whisker plot between the obtained morphological parameters and existing TAR devices (STAR, Buechel–Pappas (BP), TNK, BOX, Agility, and WSU) in Figure 6. Only a few sizes of BP and TNK were within the interquartile range (IQR) of obtained parameters, and devices like STAR and Agility showed out-of-range values for a few parameters. The dimensions of WSU TARs are out of IQR, except for anterior width values (TiW and TaW).

## 4. Discussion

In this pilot study, CT and MRI data were utilized to determine the morphological characteristics of the ankle joint. Different statistical analyses were performed on the obtained data to determine differences between the genders (females and males), age groups (young adults, adults, and older adults), and models derived from different image acquisition methods (CT and MRI). For most parameters, males showed higher mean values than females, but very few parameters showed a significant difference between males and females. This outcome is comparable with previous studies; specifically, the tibial (TiW) and talar width (TaW) values are higher in males than in females across all the sections, and the difference between the two gender groups is significant. Similar results were reported by Daud et al. (2013) and Stagni et al. (2005), where a significant difference was also observed between the genders [5,23]. Several other studies reported similar observations, where females had relatively smaller medial and later condyles, tibial length, and tibial head width compared to males [18]. Even though some of these studies used the shape variation methodology to identify differences between the genders, the observations presented are directly correlated to the reported morphological parameters that showed a significant difference between males and females in this pilot study. The gender differences observed for TaW—anterior (7 mm); TaW—posterior (6 mm); and TTL (4 mm) parameters in this study were higher compared to 4 mm for TaW—anterior; 3 mm for TaW—posterior; and 3 mm for TTL values reported by Daud et al. (2013) [23]. This can be explained by the differences in the mean age of females and males between the two studies, where the mean age of females (41 years) and males (47 years) in this study is high compared to 22 and 24 years for females and males, respectively. This mean age falls within the Group 1 mean age (23 years) of this study but with a smaller number of subjects: 5 females and 1 male compared to 49 females and 50 males in the other study. Nevertheless, when the gender differences are calculated for these parameters, the obtained values are 5 mm for TaW—anterior, 3 mm for TaW—posterior, and 6 mm for TTL. Very few studies have discussed the effect of age on ankle joint morphology. A study by Tomassoni et al. (2014) reported higher ankle length values in the older population compared to young adults and adults and observed higher ankle height and ankle circumference for adults compared to young adults and the older population [36]. Similarly, a study by Nozaki et al. (2020) reported an increase in the anteroposterior length of the trochlea and the talar head surface with aging for both females and males [25]. From the obtained results in this pilot study, we can see that the parameters (TiW, TaW, TML, and TTL) are comparable to previous study results, where these parameters show higher values with aging, with a significant difference (*p*-value < 0.05) observed between young-adults and adults for most of the parameters. The differences between these two age groups can be attributed to epiphyseal plate changes that occur in young adults and bone remodeling due to variations in mechanical stress [20]. Based on these findings, it is very clear that the critical parameters (TiW, TaW, TML, and TTL) required for the implant design show significant differences between females and males and between young adults and adults [23]. Therefore, this necessitates an implant design that can scale across genders and age groups.

The sagittal radius of the talus (SRTa) decreased from the medial to lateral section. But, no significant difference was observed between the mean values obtained in the medial (23 ± 6 mm) and middle sections (23 ± 6 mm), and a significant difference in mean values was observed between the lateral (21 ± 4) and medial sections. Similar findings were reported in other studies with SRTa values (medial—25.7 mm; middle—24.7 mm; and lateral—21.7 mm) by Siegler et al. (2014), and with a higher mid-sagittal radius value compared to the medial radius (medial—20.4 mm; middle—20.7 mm; and lateral—20.3 mm) was reported by Wiewiorski et al. (2012) [8,27]. So, based on the obtained results in this study, the talus can be modeled as a truncated cone with the apex directed towards the lateral side, therefore justifying the claims from earlier studies about the varying axis of motion [8,37,38]. The obtained talar width (TaW) values showed a wider anterior (27 mm) section compared to the posterior (21 mm) section, resulting in the shape of the trochlea tali with an apex oriented posteriorly (ATTS—12 deg). Similar results were reported in previous studies with anterior TaW (range 27–30 mm), posterior TaW (range 21–25 mm), and ATTS (9–12 deg) [6,8,23]. These values support that the cardinal system used to measure morphological parameters was successfully implemented, thereby eliminating variability between the studies.

A study by Wiewiorski et al. (2012) observed a significant difference in talar dome ratios (TDR) between the anterior and posterior sections but not between the anterior and central sections and also reported a higher dome ratio compared to values obtained in this study [27]. Riede et al. (1971) observed a higher dome ratio in the younger population (range 0.06–0.08) compared to the older population (range 0.02–0.04) [39]. From Table 1 and Table 4, we can observe that the mean age of the subjects was 44 years, which is above the age of the younger generation (18–35) years, thereby showing lower TDR values in the central (0.03) and posterior (0.03) sections and a mean TDR value of 0.04, considering an average of all the sections. In this study, we observed a higher mean value for the medial talar edge radius (R_m_—4 mm) compared to the mean lateral talar edge radius (R_l_—3 mm). A significant difference between the talar edge angles (α and β) was observed in most of the sections (see Appendix A). The mean talar edge angles (α—116 deg and β—117 deg) obtained in this study were higher compared to previous studies, and a minimal difference was observed between the mean edge angles. Previous studies showed a lateral edge angle (range 88–93 deg) and medial edge angle (range 105–113 deg) for the talus [33]. This can be explained by the 2D imaging data used in these studies to measure the morphological parameters. However, we also observed a significant difference between the acquisition methods (CT and MRI) for talar edge angles (α and β) and talar edge radius (R_l_ and R_m_) values. This may be due to magnetic field distortion by cortical bone in surrounding tissues, thereby generating geometric distortion at the interface, resulting in minor artifacts (bad edges) that might have occurred during the segmentation of the bone from the surrounding soft tissue (cartilage) [31]. There is only one study that compared CT and MRI models of the ankle joint, where they reported higher accuracy for the MRI-based bone model compared to the CT model with a 3D contour error (range 0.7–1.1 mm) between the models [40]. The obtained talar edge radius values (R_l_ and R_m_, Central and Posterior) showed a difference ranging from 1 to 4 mm between the CT and MRI models. In this pilot study, we pooled the CT and MRI models from different patients to determine the differences between the acquisition techniques rather than comparing the techniques by obtaining CT and MRI models from the same patient, and the study by Durastanti et al. (2019) used only one cadaver leg for comparison between the two imaging modalities. Several studies reported that MRI-based 3D models are comparable to their corresponding CT-based models [30,31,40]. However, these studies focused on other joints and applications. So, based on these observations, future studies should aim to acquire both CT and MRI from the same individual for a larger sample population. This will allow us to obtain accurate results by performing inter-subject and intra-subject comparisons for these techniques. Additionally, the combination of these two techniques could generate more accurate 3D models that reflect natural ankle joint morphology, thereby eliminating the segmentation artifacts seen in this pilot study [40].

A similar trend was observed in the case of tibial sagittal radius (SRTi) values, where there is a decrease in values from the medial (26 mm) to lateral section (25 mm) and similar values (range 26–29 mm) were reported in previous studies [7,15]. In the case of TiW, higher values (range 31–33 mm) were observed by Stagni et al. (2005) and Kuo et al. (2013) when compared to the values obtained in this study (range 23–27 mm) across all the sections [5,15]. Like talar width (TaW), the tibial width (TiW) values decreased from the anterior to posterior section, resembling the trochlea tali shape (ATMS—14 deg) but with an angle greater than ATTS. Compared to the mortise lengths (TML), the trochlea lengths (TTL) are higher in both the medial (TML—25 mm; TTL—35 mm) and lateral sections (TML—24 mm; TTL—32 mm), therefore confirming that the surface area of trochlea tali is greater than the surface area of the tibial mortise. When compared with other variables, the tibial (TiW) and talar width (TaW), and sagittal radius of the tibia (SRTi) and talus (SRTa) showed higher correlation coefficients (r > 0.70), and a significant relationship between them (*p*-value < 0.05). A study by Daud et al. (2013) reported a similar correlation for the talar width values (r—0.94 ± 0.04) [23]. The regression equations derived from this study help in predicting the morphological parameters of tibia based on talus dimensions and vice versa. There is only one forensic study that derived logistic regression equations to determine the sex based on the talus morphological parameters [41]. However, none of the previous studies established a significant relationship between the morphological parameters of the tibia and talus by deriving regression equations [23,42,43].

From Figure 6, we can observe that most of these devices fit only a very limited group of people, and most of them showed values out of the IQR for the tibial component parameters (TML and TiW posterior). The sagittal radius (SRTa) of these devices is out of range due to the presence of condyles; otherwise, the radius of the articulation surface is 25 mm, which lies within the IQR. These values show that the size of these devices is larger than the size required to fit 50 percent of the people in this study. When compared with previous studies, the number of specimens (*n* = 22) analyzed in this pilot study was limited and had a wide age range (13–88 years). This study did not consider height and body weight data, but studies showed that these parameters have small effects on ankle morphology, and some studies showed no correlation between the majority of morphological parameters with the patient’s body height [5,7,44]. Even though the reference cardinal system was defined based on previous studies, it is subjective and could lead to minimal changes, thereby affecting reproducibility [6]. In this study, the measurements were performed by one individual; therefore, the inter-observer reliability could not be evaluated. The coronal plane axis is defined using centers of the medial and lateral sagittal radius by excluding the center of the mid-sagittal (middle) radius, thereby affecting the morphological measurements derived from the coronal plane.

## 5. Conclusions

The pilot study used both CT and MRI data to analyze the morphological characteristics by developing 3D models of the tibia and talus. No significant difference was observed between CT and MRI models for measuring the majority of morphological parameters, but care should be taken while processing MRI data to eliminate the artifacts. The cardinal system was successfully applied to measure the morphological parameters, and the obtained results justified modeling the talus as a skewed truncated cone with its apex aligned towards the lateral side. Critical morphological parameters that are necessary to develop ankle device shows significant differences between genders and age groups. The obtained morphological parameters in this study did not fit with most of the existing TAR devices and did not fall within the IQR range for most of the parameters. The regression equations derived further established the mathematical relationship between the tibia and talus parameters, providing a guide to predict the morphology of the tibia based on talus parameters and vice versa. The obtained results from this pilot study provide some insights into data collection, 3D modeling, and the planning and designing of ankle devices that can fit a large population across genders and age groups.

## Figures and Tables

**Figure 1 bioengineering-10-01212-f001:**
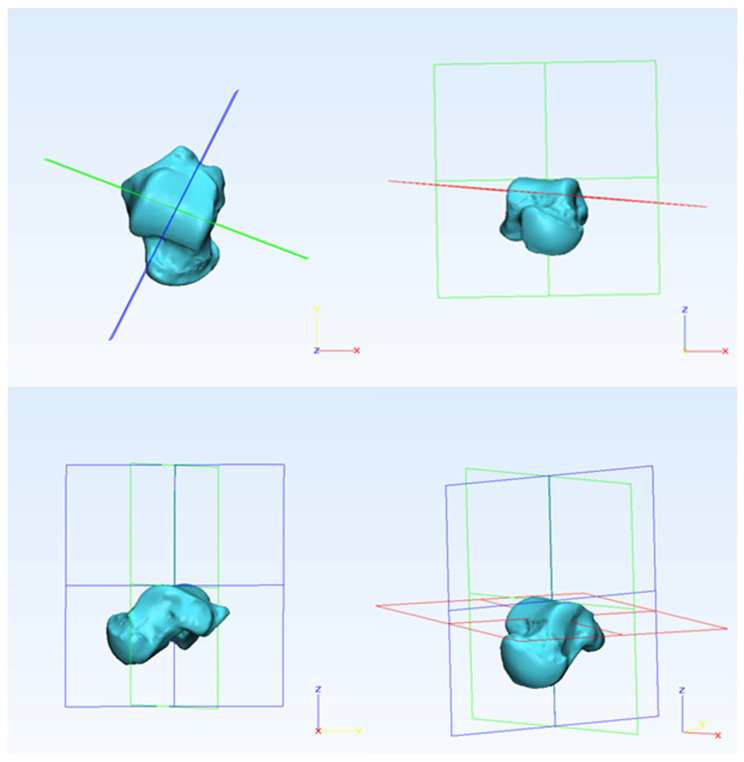
The arrangement of the reference cardinal system showing sagittal (blue), transverse (red), and coronal (green) planes.

**Figure 2 bioengineering-10-01212-f002:**
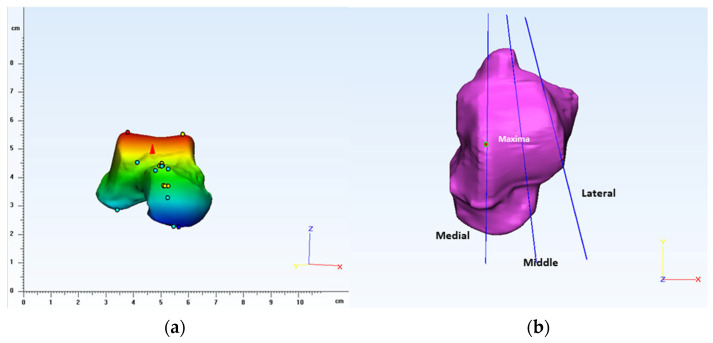
(**a**) Extrema analysis showing maximal points obtained on the talus articulation surface and (**b**) defining planes to create medial, middle, and lateral sections.

**Figure 3 bioengineering-10-01212-f003:**
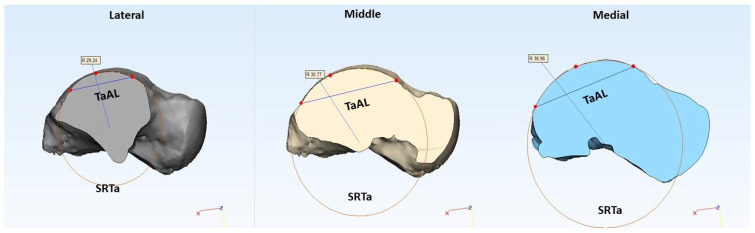
Lateral, middle, and medial sections of the talus showing corresponding morphological parameters measured in a sagittal plane.

**Figure 4 bioengineering-10-01212-f004:**
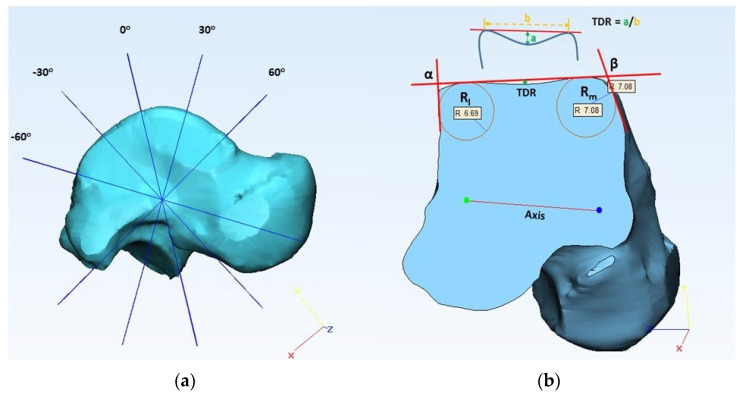
(**a**) Multiple datum planes created showing 30-degree increments to create sections in the coronal plane, (**b**) measurement of lateral talar edge radius (R_l_), medial talar edge radius (R_m_), lateral talar edge angle (α), medial talar edge angle (β), and talar dome ratio at mid-coronal section.

**Figure 5 bioengineering-10-01212-f005:**
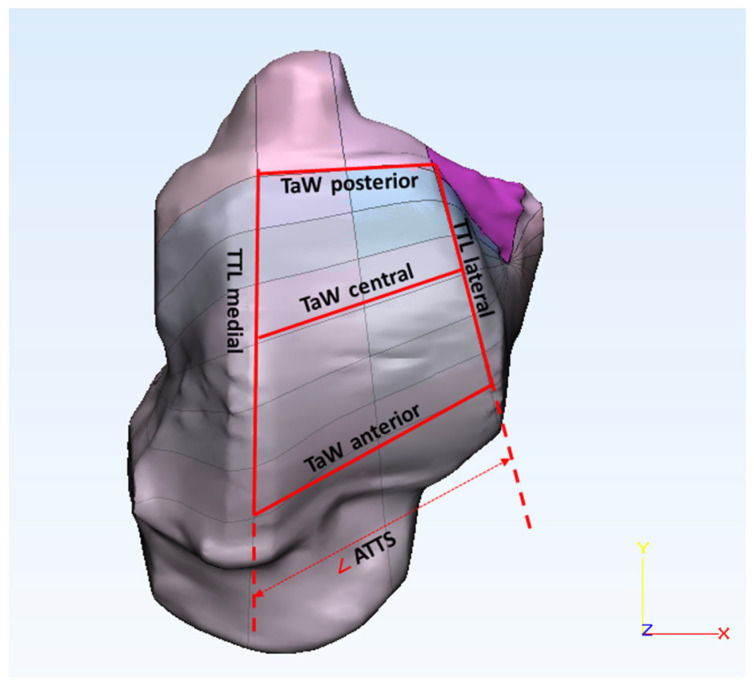
Talar dome surface showing the parameters measured: talar width (TaW—anterior, central, and posterior), trochlea tali length (TTL—medial and lateral), and angle of trochlea tali shape (ATTS).

**Figure 6 bioengineering-10-01212-f006:**
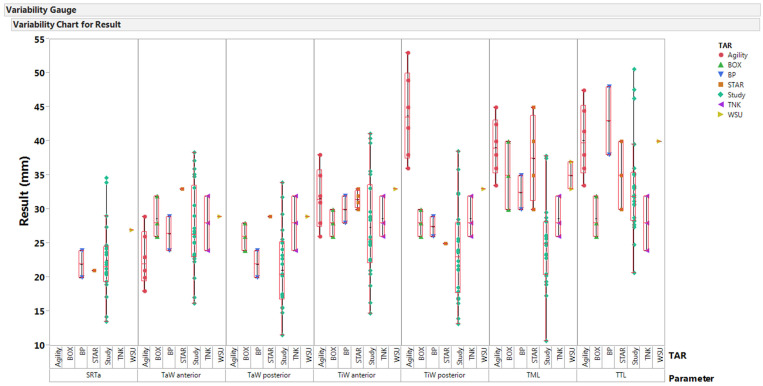
Comparison between available sizes of different TARs with morphological parameters (SRTa—trochlea tali radius; TaW—trochlea tali width; TiW—tibial width; TML—tibial mortise length; and TTL—trochlea tali length) obtained in the study [5,23].

**Table 1 bioengineering-10-01212-t001:** Demographic data of patients used in this study.

Parameters	Age					
	Total (*n* = 22)	Female (*n* = 12)	Male (*n* = 10)	Group 1 (*n* = 6) (Female = 5, Male = 1)	Group 2 (*n* = 9) (Female = 4, Male = 5)	Group 3 (*n* = 7) (Female = 3, Male = 4)
Mean	44	41	47	23	46	60
SD	17	19	13	6	3	12
Max	88	88	58	27	50	88
Min	13	19	13	13	39	52

**Table 2 bioengineering-10-01212-t002:** Summary of imaging protocols used in this study.

Imaging Protocols		
Technique	Details	
CT (General Electric, Optima 660, 64 slices) General Electric Healthcare, Milwaukee, WI, USA	The patient was placed at iso-center, the ankle positioned at 90 degrees, and the tape was used to secure the foot Slice thickness (ST)—2.5 mm with no skips Field of view (FOV)—16 cm Matrix size—512 × 512 Sagittal and coronal reconstructions—0.625 mm With bone and soft tissue windows	
MRI 1.5 Tesla scanner (General Electric, Optima 450 W) General Electric Healthcare, Milwaukee, WI, USA	INVIVO/GE 1.5 T HD 8 ch foot/ankle coil is used to maintain ankle position at 90 degrees Matrix size—256 × 192 Number of excitations (NEX)—2 Bandwidth (BW)—31.25 kHz	
	Axial	T1 and T2 weighted, fat-saturated, Fast spin-echo (FSE) sequence FOV—12 cm ST—3 mm skip 1
	Coronal	T2 weighted, fat-saturated, FSE sequence FOV—14 cm ST—3 mm skip 1
	Sagittal	T1 weighted (SE), Short tau inversion recovery (STIR) sequence FOV—14 cm ST—4 mm skip 0.5

**Table 3 bioengineering-10-01212-t003:** List of morphological parameters measured in different sections and their definitions [7,27,33].

Variable	Section	Definition
Tibia parameters		
TiAL (mm)	(medial, middle, lateral)	Tibial arc length
SRTi (mm)	(medial, middle, lateral)	Tibial sagittal radius
TiW (mm)	(anterior, central, posterior)	Tibial width
TML (mm)	(medial, lateral)	Tibial mortise length
ATMS (deg)	-	Angle of tibial mortise shape
Talus parameters		
TaAL (mm)	(medial, middle, lateral)	Trochlea tali arc length
SRTa (mm)	(medial, middle, lateral)	Trochlea tali radius
TaW (mm)	(anterior, central, posterior)	Trochlea tali width
TTL (mm)	(medial, lateral)	Trochlea tali length
ATTS (deg)	-	Angle of trochlea tali shape
TDR	(anterior, central, posterior)	Talus dome ratio
α (deg)	(anterior, central, posterior)	Lateral talar edge angle
β (deg)	(anterior, central, posterior)	Medial talar edge angle
R_l_ (mm)	(anterior, central, posterior)	Lateral frontal talar edge radius
R_m_ (mm)	(anterior, central, posterior)	Medial frontal talar edge radius

**Table 4 bioengineering-10-01212-t004:** List of equations relating tibial and talar morphological parameters in different sections.

Talus Parameter	Tibia Parameter	Section	Equation	*p*-Value	R-Squared
SRTa	SRTi	Medial	SRTi = −1.987 + 1.213 × SRTa	<0.0001	0.723
		Middle	SRTi = 0.415 + 1.118 × SRTa	<0.0001	0.913
		Lateral	SRTi = 6.348 + 0.885 × SRTa	<0.0001	0.540
TaW	TiW	Anterior	TiW = 2.259 + 0.925 × TaW	<0.0001	0.608
		Central	TiW = 3.907 + 0.9 × TaW	<0.0001	0.610
		Posterior	TiW = 6.175 + 0.822 × TaW	<0.0001	0.525
TTL	TML	Medial	TML = −6.747 + 0.905 × TTL	<0.0001	0.848
		Lateral	TML = 11.433 + 0.411 × TTL	0.0036	0.352
ATMS	ATTS	-	ATMS = 5.337 + 0.731 × ATTS	0.0269	0.222

## Data Availability

The data presented in this study are available in the main sections and under the Appendix A of this article.

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
