# Peer review of "Anthropomorphic Characterization of Ankle Joint"

_bioengineering, 2023, doi:10.3390/bioengineering10101212_

Round 1

Reviewer 1 Report

Dear Authors,

I read the article on the anthropometric characterization of the ankle joint with interest.

I consider the detailed description of the joint and the illustrated definition of the planes, distances, and angles used for measurement to be a definite result.

The weakness of the research plan is that the two technologies were compared in a separate patient group, i.e., the study was based on eight CT and 14 MRI scans. The 22 subjects should be sufficient if both measurements had taken place in each case and the conclusions had been drawn based on corresponding samples. Unfortunately, we do not know the composition of the two samples at such a level that this result would justify the statements in the conclusion.

In addition to supplementing the test sample, it is necessary to name the performed statistical tests precisely each time because, in this article, it is not clear which tests would prove the claims with which results.

I miss a more precise presentation of the examined persons, since in addition to the anthropometric characteristics, and presumably also in the case of the ankle prosthesis, in addition to age and gender, the features of the population (for example, Chinese), body size, body structure, the stronger/weaker or left/right leg, the foot shape or size can have a fundamental effect.

I recommend the imaging protocols section in a separate table by separating Table 1.

Based on the above, I recommend revising the article.

I suggest refining the wording in the abstract and introduction, presenting views contradicting the simplified statements presented.

Author Response

The authors would like to thank the reviewers for their valuable feedback, which helped us to improve the quality of the manuscript. Please see the responses below for each of the reviewer’s comments. All the changes made to the article are highlighted in yellow. A few spelling and grammatical corrections are made in the manuscript. We would be glad to respond to any further questions or comments that you may have.

Reviewer 1 Comments:

Dear Authors,

I read the article on the anthropometric characterization of the ankle joint with interest. I consider the detailed description of the joint and the illustrated definition of the planes, distances, and angles used for measurement to be a definite result. The weakness of the research plan is that the two technologies were compared in a separate patient group, i.e., the study was based on eight CT and 14 MRI scans. The 22 subjects should be sufficient if both measurements had taken place in each case and the conclusions had been drawn based on corresponding samples. Unfortunately, we do not know the composition of the two samples at such a level that this result would justify the statements in the conclusion.

Response: Thank you for your valuable input. Since this is a pilot study with fewer subjects, our primary focus was to understand the effect of gender on the ankle joint morphology and to establish a relationship between tibia and talus parameters. This study provided us with some insights regarding data collection, methods used, and how they affected the results. These observations and the reviewer’s feedback will be considered in an extended study, thereby using the results to develop and optimize the TAR devices.

In addition to supplementing the test sample, it is necessary to name the performed statistical tests precisely each time because, in this article, it is not clear which tests would prove the claims with which results.

Response: The statistical methods used in this study are now elaborated. Please see the lines 211-229 in page 8 for the updates. The methods are now listed next to the observed results and claims they support. Please see the lines 99-107 in pages 2 and3, lines 241-251 in page 8, lines 275-306 in page 11 for details.

I miss a more precise presentation of the examined persons, since in addition to the anthropometric characteristics, and presumably also in the case of the ankle prosthesis, in addition to age and gender, the features of the population (for example, Chinese), body size, body structure, the stronger/weaker or left/right leg, the foot shape or size can have a fundamental effect.

Response: We apologize, we do not have any other information except age and gender for this pilot study necessitated by the ethical review board. The age groups are now included and compared in the revised article.

I recommend the imaging protocols section in a separate table by separating Table 1.

Response: The imaging protocols section is removed from Table 1, and now presented as Table 2.

Based on the above, I recommend revising the article. I suggest refining the wording in the abstract and introduction, presenting views contradicting the simplified statements presented.

Response: The wording in the abstract and introduction has been revised to properly reflect the results of the article. Please see the highlighted sections of the abstract (lines 13-26) and introduction (57-73, 94-110) for the changes.

Reviewer 2 Report

Thanking the authors for the work done, I do not recommend the publication of the article as the scientific level of the manuscript is not sufficient to justify its publication in this journal.

Moderate editing of the language is needed

Author Response

The authors would like to thank the reviewers for their valuable feedback, which helped us to improve the quality of the manuscript. Please see the responses below for each of the reviewer’s comments. All the changes made to the article are highlighted in yellow. A few spelling and grammatical corrections are made in the manuscript. We would be glad to respond to any further questions or comments that you may have.

Reviewer 2 Comments:

Thanking the authors for the work done, I do not recommend the publication of the article as the scientific level of the manuscript is not sufficient to justify its publication in this journal.

Response:  We missed the opportunity to revise the manuscript with your comments. We have made numerous edits to the manuscript to improve the quality based on the feedback from the reviewers contained in this report.

Reviewer 3 Report

1)  The last paragraph of the conclusion states that the motivation for the study was to measure 15 morphological parameters and to test the hypotheses of significant differences between males and females, similarity between CT and MRI models and significant relationships between tibia and talus parameters.   The above points need more clarification. Reference 7, for example, claims 31 measured parameters which calls into question the statement on line 73. Differences between males and females should be expected for a weight-bearing joint, but the discussion in the first paragraph of section 4 doesn't seem to arrive at any conclusion different from what has already been reported in references 5 and 18.   The only discussion with respect to comparison between CT and MRI is on line 325 for acquisition methods. Some perspective could perhaps be added by including relevant information from other studies such as G Durastanti et al., Comparison of cartilage and bone morphological models of the ankle joint derived from different medical imaging technologies, Quantitative Imaging in Medicine and Surgery, 9 (2019) 1368-1382.   There is no discussion in section 4 concerning the relationship between tibia and talus with respect to other studies. Information in N Tumer et al., Three-dimensional analysis of shape variations and symmetry of the fibula, tibia, calcaneus and talus, Journal of Anatomy (2018) may be of interest.   2)  Unless there is a specific reason for doing so, it seems unnecessary to report data to two decimal places. Table 3 and the paragraph between lines 213-234 are a sea of numbers and would be less intimidating for readers if values were rounded off to the nearest whole number. There were only 22 subjects in the study and reporting to two decimal places doesn't make the results more accurate.   3)  The last sentence in the conclusion repeats what has been concluded in other studies, for example reference 6, that 3D models can be used for fabrication of individual ankle replacement devices. But this seems to be mostly an economic consideration, cost of an off the shelf prosthesis as compared to fabrication according to patient specifications. A few sentences about the prospects for the latter would be useful.   4)  There are three figure captions on the last page of the supplementary data, but no figures. The two correlation plots seem to require that the reader mentally rotates the horizontal axis down along the side of the figure to obtain the vertical axis and this should probably be explicitly stated in the captions.  

Author Response

The authors would like to thank the reviewers for their valuable feedback, which helped us to improve the quality of the manuscript. Please see the responses below for each of the reviewer’s comments. All the changes made to the article are highlighted in yellow. A few spelling and grammatical corrections are made in the manuscript. We would be glad to respond to any further questions or comments that you may have.

Reviewer 3 Comments:

  • The last paragraph of the conclusion states that the motivation for the study was to measure 15 morphological parameters and to test the hypotheses of significant differences between males and females, similarity between CT and MRI models and significant relationships between tibia and talus parameters.   The above points need more clarification. Reference 7, for example, claims 31 measured parameters which calls into question the statement on line 73. Differences between males and females should be expected for a weight-bearing joint, but the discussion in the first paragraph of section 4 doesn't seem to arrive at any conclusion different from what has already been reported in references 5 and 18. The only discussion with respect to comparison between CT and MRI is on line 325 for acquisition methods. Some perspective could perhaps be added by including relevant information from other studies such as G Durastanti et al., Comparison of cartilage and bone morphological models of the ankle joint derived from different medical imaging technologies, Quantitative Imaging in Medicine and Surgery, 9 (2019) 1368-1382. There is no discussion in section 4 concerning the relationship between tibia and talus with respect to other studies. Information in N Tumer et al., Three-dimensional analysis of shape variations and symmetry of the fibula, tibia, calcaneus and talus, Journal of Anatomy (2018) may be of interest.  

Response: Thank you for providing your observations and reference articles to improve the quality of this manuscript. For clarification, the study in reference 7 reported 31 parameters measured across all the cardinal planes. In this study, we reported a total of 39 parameters (excluding age) measured across these planes for the 15 main morphological parameters. The differences observed between the genders, and image acquisition techniques are now elaborated by comparing with the results from previous studies. Please see page 8, lines 241-251, and page 14, lines 336-376 for details. Unfortunately, the referenced article (N Tumer et al. 2018) does not provide any discussion regarding the relationship between the tibia and talus but helped in discussing the gender differences. We have explored other studies discussing the correlation; however, those studies did not determine any relationship between the tibia and talus morphological parameters.

  • Unless there is a specific reason for doing so, it seems unnecessary to report data to two decimal places. Table 3 and the paragraph between lines 213-234 are a sea of numbers and would be less intimidating for readers if values were rounded off to the nearest whole number. There were only 22 subjects in the study and reporting to two decimal places doesn't make the results more accurate.  

Response: The values are rounded wherever possible to the nearest whole number, in Table 3 and updated accordingly in corresponding text sections.

  • The last sentence in the conclusion repeats what has been concluded in other studies, for example reference 6, that 3D models can be used for fabrication of individual ankle replacement devices. But this seems to be mostly an economic consideration, cost of an off-the-shelf prosthesis as compared to fabrication according to patient specifications. A few sentences about the prospects for the latter would be useful.  

  Response: Thank you. The focus of this study was to establish a relationship between tibial and talar morphological parameters. This relationship was a key parameter needed to scale the implant sizes that can fit a wider population across genders rather than the currently available off-the-shelf prostheses that fit a limited population (less than 50% in this study). The wording in the conclusion is now revised to properly reflect the findings of the article.

  • There are three figure captions on the last page of the supplementary data, but no figures. The two correlation plots seem to require that the reader mentally rotates the horizontal axis down along the side of the figure to obtain the vertical axis and this should probably be explicitly stated in the captions.  

Response: There are three regression plots associated with figure captions on the last page of the supplementary data. We have uploaded a pdf version of the file. For both the color maps, the vertical axis with parameters is now added to facilitate readers.

Reviewer 4 Report

BIOENGINEERING-2587353 presents results for ankle joint anthropomorphic characterization. While some parts of the paper were interesting, other areas could be improved. I hope the authors consider my feedback.

MAJOR COMMENTS

·         Table 1: The age demographic is large. Is it ok to compare young adults to older adults?

·         Are there any other demographic characteristics that can be included to help describe the sample?

·         Statistical Analysis: Justification needs to be provided as to why t-tests were conducted by sex. Might age strata be more interesting?

MINOR COMMENTS

·         Lines 76-79: Revise to just be a clear purpose statement.

·         The information in Table 1 can be integrated into the text. Age can also be presented as mean+-SD, given that min and max are already measures of variability.

·         The authors need to list why in the manuscript text IRB approvals were not necessary for this study.  

·         Figure 6 probably belongs in the Results section.

·         Make any changes to the abstract that align with those from the text.

Author Response

The authors would like to thank the reviewers for their valuable feedback, which helped us to improve the quality of the manuscript. Please see the responses below for each of the reviewer’s comments. All the changes made to the article are highlighted in yellow. A few spelling and grammatical corrections are made in the manuscript. We would be glad to respond to any further questions or comments that you may have.

Reviewer 4 Comments:

BIOENGINEERING-2587353 presents results for ankle joint anthropomorphic characterization. While some parts of the paper were interesting, other areas could be improved. I hope the authors consider my feedback.

MAJOR COMMENTS

  • Table 1: The age demographic is large. Is it ok to compare young adults to older adults?

Response: We appreciate your feedback. The population is now divided into three groups (Group 1: below 30 years, Group 2: between 30-50 years and Group 3: above 50 years). We decided to divide the population into 3 to get the distribution count equally close between the groups rather than with a huge difference. Please see page 3, lines 113-120 for the updates. Our thought process was to include all since it was a pilot study with limited subjects.

  • Are there any other demographic characteristics that can be included to help describe the sample?

Response: We were limited by the ethical review board and federal regulations regarding subject details that we could not bring to our analysis.

  • Statistical Analysis: Justification needs to be provided as to why t-tests were conducted by sex. Might age strata be more interesting?

Response: There are not many studies that explored the effect of age on ankle joint morphology and a few studies (Anderson et al., 1956, Stagni et al., 2005) have concluded that age has shown small effects on ankle morphology in the adult population. So, age groups were not included initially in our study and no t-tests were conducted. Thanks for your valuable feedback; we identified some new findings about the effect of age on ankle joint morphology. The manuscript was accordingly revised to include age groups and t-tests results by age group and the obtained results are discussed in the article. Please see page 8, lines 243-247, page 14, lines 354-376, and table 4 for the updates.  

MINOR COMMENTS

  • Lines 76-79: Revise to just be a clear purpose statement.

Response: The statements have been revised to properly reflect the purpose of the study. Please see pages 2 and 3, lines 94-110 for the updates.

  • The information in Table 1 can be integrated into the text. Age can also be presented as mean +SD, given that min and max are already measures of variability.

Response: The patient demographics are integrated into the text. Please see page 3, lines 113-116 for details.

  • The authors need to list why in the manuscript text IRB approvals were not necessary for this study.

Response: The IRB approval statement has been updated. It was included under the acknowledgments section initially. Please see page 17, lines 499-501 for the updates.

  • Figure 6 probably belongs in the Results section.

Response: Figure 6 and corresponding details are moved to the results section. Details of the plot are described in the methods section. Please see page 12, lines 325-331 for the updates.

  • Make any changes to the abstract that align with those from the text.

Response: The wording in the abstract is now revised to properly reflect the results of the article. 

Reviewer 5 Report

Dear Authors, 

The paper is very well written. In my opinion, it does not need any corrections. The only comment I have is about figure 6. It should be bigger, because in this form one cannot see anything. It can be extended to the page and the font should be bigger. In addition, it is worth explaining the abbreviations in the description of this figure. I mark acceptance in this form, but please improve the appearance of the figure.

Author Response

The authors would like to thank the reviewers for their valuable feedback, which helped us to improve the quality of the manuscript. Please see the responses below for each of the reviewer’s comments. All the changes made to the article are highlighted in yellow. A few spelling and grammatical corrections are made in the manuscript. We would be glad to respond to any further questions or comments that you may have.

Reviewer 5 Comments:

Dear Authors, 

The paper is very well written. In my opinion, it does not need any corrections. The only comment I have is about figure 6. It should be bigger, because in this form one cannot see anything. It can be extended to the page and the font should be bigger. In addition, it is worth explaining the abbreviations in the description of this figure. I mark acceptance in this form, but please improve the appearance of the figure.

Response: We appreciate your comments. The figure has been enhanced to a larger size to fit within the page layout and parameter abbreviations are explained in the description. A bigger size figure is uploaded separately which can be used during the publication process. Please see page 13 for the re-sized figure and title.

Round 2

Reviewer 1 Report

Thank you for the improvement.

You should move Table 4. to the annexe.

Hope you will have a larger sample, and collect both an MRI and CT for your subjects.

Bests  

Author Response

The authors would like to thank the reviewers for accepting the revisions made to the first version of the article. Please see the responses below for the minor comments provided during round 2. All the changes made to the article are highlighted in yellow. We hope the latest version of the manuscript addressed all the remaining concerns and looking forward to hearing your decision soon.

Reviewer 1 Comments:

Thank you for the improvement.

Response: Thank you. We are glad that we have addressed all your major comments and hope that we were able to respond to your remaining comments in this version of the manuscript.

You should move Table 4. to the annexe.

Response: The table is moved to the Appendix A section of the manuscript and appropriately cited in the text.

Hope you will have a larger sample, and collect both an MRI and CT for your subjects.

Response: Thank you for your valuable input. Since this is a pilot study our scope is limited with fewer subjects. The observations from the present study and the reviewer’s feedback to collect both MRI and CT for a larger population will be considered in our extended study.

Reviewer 4 Report

The authors have addressed my previous critiques but here is some additional feedback for the authors:

***Lines 17-20: Statistics need to be included in these statements to support the text. I understand word count, but revise accordingly to include. 

***Line 21: "(r->0.90)" needs correction. 

***Lines 99-106: Perhaps this was a comment from another reviewer, but it is odd to have information about stat analysis in the Introduction. This information belongs in the Methods. 

***Lines 117-120: Please justify in the text why this age grouping was selected. Also not sure if we can suggest persons aged 50 years are "old". 

***Lines 451-460: Remove results from a discussion section and avoid re-introducing data elements for disrupting prospective flow (Table 5). 

Author Response

The authors would like to thank the reviewers for accepting the revisions made to the first version of the article. Please see the responses below for the minor comments provided during round 2. All the changes made to the article are highlighted in yellow. We hope the latest version of the manuscript addressed all the remaining concerns and looking forward to hearing your decision soon.

Reviewer 4 Comments:

The authors have addressed my previous critiques but here is some additional feedback for the authors:

Response: Thank you. We are glad that we have addressed all your major comments and hope that we were able to respond to your remaining comments in this version of the manuscript.

***Lines 17-20: Statistics need to be included in these statements to support the text. I understand word count, but revise accordingly to include. 

Response: Thanks for your comment. The p-value and parameters are included to support the text. Please see (lines 17-24) for details.

***Line 21: "(r->0.90)" needs correction. 

Response: Sorry for the confusion. For a few parameter relationships, the correlation is r >0.90. Considering all the tibia and talus width and sagittal radius parameter relationships, the correlation range (r = 0.73 – 0.97) was observed. So, the value has been updated to (r > 0.70).

***Lines 99-106: Perhaps this was a comment from another reviewer, but it is odd to have information about stat analysis in the Introduction. This information belongs in the Methods. 

Response: Thanks for your comment. Yes, we included this information in the introduction section per the request of one of the reviewers. This information is explained in the methods section (lines 208-223). So, it has been removed from the Introduction section.

***Lines 117-120: Please justify in the text why this age grouping was selected. Also not sure if we can suggest persons aged 50 years are "old". 

Response: We appreciate your feedback. The justification is now added in the manuscript text (lines 112-114). We now changed the group name to “older adults” for persons aged above 50 years.

***Lines 451-460: Remove results from a discussion section and avoid re-introducing data elements for disrupting prospective flow (Table 5). 

Response: Thanks for pointing this out. The result values and data elements disrupting the flow of the discussion were removed. The text is moved to align with the results presented in previous sections. Please see (lines 440-446) for details.